# LONG-TERM PLANNING, SHORT-TERM ADJUSTMENTS

## ABSTRACT

Deep reinforcement learning (RL) algorithms can learn complex policies to optimize agent operation over time. RL algorithms have shown promising results in solving complicated problems in recent years. However, their application on real-world physical systems remains limited. Despite the advancements in RL algorithms, the industries often prefer traditional control strategies. Traditional methods are simple, computationally efficient and easy to adjust. In this paper, we propose a new Q-learning algorithm for continuous action space, which can bridge the control and RL algorithms and bring us the best of both worlds. Our method can learn complex policies to achieve long-term goals and at the same time it can be easily adjusted to address short-term requirements without retraining. We achieve this by modeling both short-term and long-term prediction models. The short-term prediction model represents the estimation of the system dynamic while the long-term prediction model represents the Q-value. The case studies demonstrate that our proposed method can achieve short-term and long-term goals without complex reward functions.

## 1 INTRODUCTION

Optimal control methodologies use system dynamic equations to design actions that minimize desired cost functions. A cost function can be designed to track a trajectory, reach a goal, or avoid obstacles. It is also possible to design a cost function to achieve a combination of goals. Model Predictive Control (MPC) is a common optimal control technique and has been applied to many industrial applications such as pressure control and temperature control in chemical processes (Garcia et al., 1989). The traditional control solutions are not adequate to address the challenges raised with the evolution of industrial systems. Recently, deep reinforcement learning (RL) has shown promising results in solving complex problems. For example, it has generated superhuman performance in chess and shogi (Silver et al., 2017). The following advantages make deep RL a strong candidate to overcome traditional control limitations. First, deep RL has an advantage in solving complex problems, especially when the consequences of an action are not immediately obvious. Moreover, it can learn an optimal solution without requiring detailed knowledge of the systems or their engineering designs. Finally, deep RL is not limited to time-series sensors and can use new sensors such as vision for a better control.

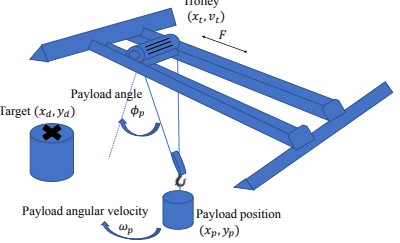

Figure 1: Crane system. The long-term goal is to move the payload to the target location, $(x_d, y_d)$ as soon as possible. The short-term goal is to have zero sway at the destination, $\omega_p = 0$.

However, deep RL has not been applied to address industrial problems in a meaningful way. There are several key issues that limit the application of deep RL to real-world problems. Deep RL al-

gorithms typically require many samples during training (sample complexity). Sample complexity leads to high computational costs. A high computational cost can be justified for industries as a one-time charge. However, oftentimes small changes in the system goal, such as changing the desired temperature in a chemical reactor, or a new constraint such as a maximum allowable temperature in the reactor, require retraining the model. Moreover, industrial systems often have several short-term and long-term objectives. For example, consider the crane system shown in Figure 1. The long-term goal is to convey the payload to the target location as soon as possible. However, when the payload gets close to the target, it must have minimum sway for the safety of the operators. Designing a reward function that can capture these short-term and long-term goals concurrently can be challenging or even infeasible.

A class of short-term objectives related to safe exploration during RL training have been studied recently. Gu et al. (2017) presented an application of deep RL for robotic manipulation control. To ensure safe exploration, they set maximum limits for the joint positions, and joint velocities. Moreover, they set a sphere boundary for the end-effector position and when the boundaries were about to be violated, they used correction velocity to force the end-effector position back to the center of the sphere. Dalal et al. (2018) formulated the safe exploration as an optimization problem. They proposed to add a safety layer that modifies the action at each time step. Toward this end, they learn the constraint function using a linear model and use this model to find the minimal change to the action such that the safety constraints are met at each time step. To the best of our knowledge, there is no study addressing short-term objectives during application.

In this paper, we present a Locally Linear Q-Learning (LLQL) algorithm for continuous action space. The LLQL includes a short-term prediction model, a long-term prediction model, and a controller. The short-term prediction model represents a locally linear model of the dynamic system, while the long-term prediction model represents the value function + a locally linear advantage function. The controller uses the short-term prediction model and the long-term prediction model to generate actions that achieve short-term and long-term goals simultaneously. It adopts a policy that maximizes Q-value while achieving short-term goals. The LLQL algorithm has the following advantages:

- It does not require designing sensitive reward functions for achieving short-term and long-term goals concurrently.
- It shows better performance in achieving short-term and long-term goals compared to the traditional reward modification methods.
- It is possible to modify the short-term goals without time-consuming retraining.

The remainder of this paper is organized as follows. Section 2 represents the background in dynamic systems and RL. Section 3 represents the LLQL algorithm. Section 4 presents our methodology to achieve short-term and long-term goals using LLQL. Section 5 presents our experimental results. Section 6 presents the conclusions of the paper. Section A discusses the related work. Section B presents additional experiments for those interested.

## 2 BACKGROUND AND DEFINITIONS

In this section, we review the backgrounds in dynamic systems and reinforcement learning.

### 2.1 DYNAMIC SYSTEMS

A continuous-time dynamic system can be represented as:

$$\frac{dx(t)}{dt} = f(x(t), u(t), t; p), \tag{1}$$

where given the system parameters, $p$, $f$ maps the state variables, $x \in X$, and actions, $u \in U$, to the state derivative, $\frac{dx}{dt}$ at time $t$. In state space control, the goal is to design a control policy, $\pi_{control}(u(t)|x(t))$, that generates proper actions so as the state variables follow the given desired trajectory, $x_d(t)$. It is challenging to design a control policy for a nonlinear complex system represented in equation (1).

The control problem becomes much easier to address when this system is linear with respect to the input (Chen et al., 2003). We can present these systems as:

$$\frac{dx(t)}{dt} = f(x(t)) + g(x(t))u(t). \tag{2}$$

Since measurements are typically sampled in discrete times, we derive a discrete time version of linear system (2). Using a first-order approximation:

$$\frac{dx(t_k)}{dt} = \frac{x(t_{k+1}) - x(t_k)}{t_{k+1} - t_k}, \tag{3}$$

where $t_k$ represents time at sample point $k$. In this paper, we assume the sampling rate is constant; $\Delta = t_{k+1} - t_k$. Using (2) and (3), we have:

$$x(t_{k+1}) - x(t_k) = \Delta(f(x(t_k)) + g(x(t_k))u(t_k)). \tag{4}$$

For brevity, we present $t_k$ by $k$, $f(x(t_k))$ by $f(x_k)$, and $g(x(t_k))$ by $g(x_k)$ in the remainder of the paper. Therefore, we can represent a dynamic system as:

$$x_{k+1} = x_k + \Delta(f(x_k) + g(x_k)u_k). \tag{5}$$

## 2.2 REINFORCEMENT LEARNING

The goal of RL is to learn a policy, $\pi_{RL}(u_k|x_k)$, that generates a set of actions, $u \in U$, that maximize the expected sum of rewards in the environment, $E_n$. Consider:

$$R_k = \sum_{i=k}^{T} \gamma^{i-k} r(x_i, u_i), \tag{6}$$

where $\gamma < 1$ is the discount factor, $r$ is the reward function and $T$ represents the end time and can be set to $T = \infty$. The goal is to learn $\pi_{RL}$ for environment, $E_n$, such that:

$$\max(R = \mathbb{E}_{r_{i \geq 1}, x_{i \geq 1} \sim E_n, u_{i \geq 1} \sim \pi_{RL}}[R_1]). \tag{7}$$

Unlike control algorithms, model-free reinforcement learning algorithms assume the system dynamic is unknown. Q-function, $Q^{\pi}(x_k, u_k)$ is defined as the expected return at state $x_k$ when we take action $u_k$ and adopt policy $\pi$ afterward:

$$Q^{\pi}(x_k, u_k) = \mathbb{E}_{r_{i \geq k}, x_{i \geq k} \sim E_n, u_{i \geq k} \sim \pi}[R_k|x_k, u_k]). \tag{8}$$

Q-learning algorithms (Watkins & Dayan, 1992) are among the most common model-free RL methods for discrete action space problems. These algorithms use the Bellman recursive equation to model Q-function:

$$Q^{\mu}(x_k, u_k) =$$
$$\mathbb{E}_{r_{i \geq k}, x_{i > k} \sim E_n}[r(x_k, u_k) + \gamma Q^{\mu}(x_{k+1}, \mu(x_{k+1}))]), \tag{9}$$

where $\mu$ represents a greedy deterministic policy that selects the action which maximizes Q-value at each step:

$$\mu(x_k) = \text{argmax}_u Q(x_k, u_k). \tag{10}$$

Q-learning algorithms learn the parameters of the function approximator, $\theta^Q$, by minimizing the Bellman error:

$$\min(L(\theta^Q) = \mathbb{E}_{r_k, x_k \sim E_n, u_k \sim \beta}[(Q(x_k, u_k|\theta^Q) - y_k)^2]),$$
$$y_k = r(x_k, u_k) + \gamma Q(x_{k+1}, \mu(x_{k+1})), \tag{11}$$

where $y_k$ is the fixed target Q-function, and $\beta$ represents the exploration policy.

For continuous action domain problems, it is not trivial to solve equation (10) at each time step. Finding an action to maximize $Q$ which can be a complex nonlinear function is computationally expensive or even infeasible. To address this problem, Lillicrap et al. (2015) proposed the Deep Deterministic Policy Gradient (DDPG) algorithm, which learns two networks simultaneously. The critic network learns Q-function by minimizing the Bellman error, and the actor network learns parameters of the policy to maximize the estimated value of Q-function. Gu et al. (2016) proposed

Normalized Advantage Function (NAF) Q-learning which formulates the Q-function as the sum of the value function, $V(x)$, and the advantage function, $A(x, u)$.

$$Q(x, u|\theta^Q) = V(x|\theta^V) + A(x, u|\theta^A), \tag{12}$$

where

$$A(x, u|\theta^A) = -\frac{1}{2}(u - \mu(x|\theta^u))^T P(x|\theta^P)(u - \mu(x|\theta^u)). \tag{13}$$

$P(x|\theta^P) = L(x|\theta^P)L(x|\theta^P)^T$, where $L(x|\theta^P)$ is a lower-triangular matrix. The value function is not a function of action, $u$. Therefore, the action which maximizes advantage function, $A$, maximizes the $Q$ function. $P(x|\theta^P)$ is a positive-definite matrix, and therefore, the action that maximizes the advantage function and the $Q$-function is given by $\mu(x|\theta^u)$.

# 3 LOCALLY LINEAR Q-LEARNING

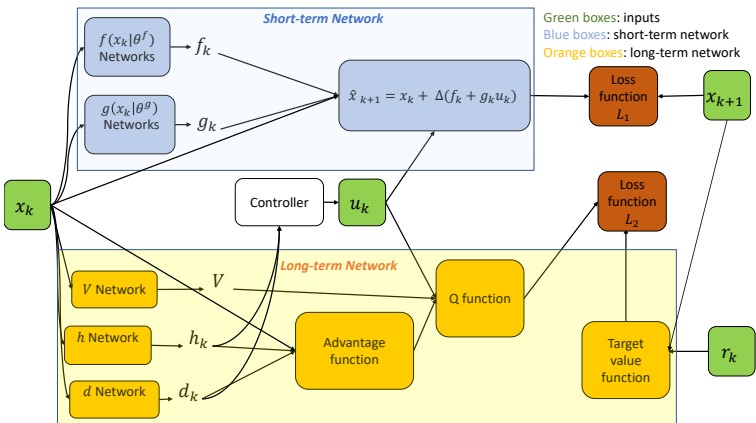

Figure 2: Learning the LLQL Network Parameters.

In this section, we propose the LLQL algorithm, which like (Lillicrap et al., 2015) and (Gu et al., 2016) can handle continuous action space. Our approach learns short-term and long-term prediction models. Using the long-term and short-term models, a controller generates actions that guide the system toward its short-term and long-term goals. Figure 2 shows our proposed structure to learn the parameters of the short-term and long-term prediction models.

*Short-term prediction:* consider the nonlinear system presented in equation (5). In this work, we use deep neural networks to estimate system functions, $f(x_k)$, and $g(x_k)$ at each operating point. Substituting the network estimations for these functions in equation (5), we can predict the next state as:

$$\hat{x}_{k+1} = x_k + \Delta(f(x_k|\theta^f) + g(x_k|\theta^g)u_k), \tag{14}$$

where $\hat{x}_{k+1}$ represents our estimation of the next step, and $\theta^f$ and $\theta^g$ are the network parameters. $\Delta$ is a constant hyper parameter. In dynamic systems, the difference between two consecutive states, $x_{k+1} - x_k$, is typically very small. Considering a small $\Delta$ leads to reasonable $f$ and $g$ values and, therefore, improves learning time and accuracy.

We call this dynamic system model *short-term prediction model*. The controller uses this model to generate actions, which lead the system toward its short-term goals. Note that previous work have used the system short-term dynamic model for generating additional samples in imagination rollout (for example, see (Gu et al., 2016), and (Racanière et al., 2017)). In this paper, we show that this model can also be used to design actions to achieve short-term goals. To learn the parameters of our short-term prediction model, $\theta^f$ and $\theta^g$, we minimize the short-term loss function, $L_1$, as it is presented in Algorithm 1.

*Long-term prediction:* Q-function represents the maximum cumulative reward that can be achieved from current state, $x_k$, taking an action $u_k$. Therefore, by learning Q-function, we learn the *long-term prediction model* for the system. Like NAF (Gu et al., 2016) (see equation (12)), we present

Q-function as a sum of value function and advantage function. However, we present the advantage function, $A(x, u|\theta^A)$ using a locally linear function of $x_k$ and $u_k$ as:

$$Q(x, u|\theta^Q) = V(x|\theta^V) + A(x, u|\theta^A),$$
$$A(x, u|\theta^A) = -||(h(x_k|\theta^h) + d(x_k|\theta^d)u_k)||, \tag{15}$$

where $h(x_k|\theta^h)$ and $d(x_k|\theta^d)$ networks model the locally linear advantage function. Note the NAF advantage function is a special case of the LLQL advantage function when $d(x_k|\theta^d) = I$, where $I$ represents the identity matrix.

To maximize Q-function, we have to design $u_k$ which minimizes $h(x_k|\theta^h) + d(x_k|\theta^d)u_k$. For simplicity, we present $h(x_k|\theta^h)$, and $d(x_k|\theta^d)$ with $h_k$ and $d_k$ respectively in the remainder of the paper. To maximize Q-function and achieve the long-term goal, we can use simple pseudo-inverse matrix multiplication and derive a solution with the least squares error as:

$$u_k = -(d_k^T d_k)^{-1} d_k^T h_k. \tag{16}$$

When $||d_k|| = 0$, it means the network predicts that our action has no impact on the advantage function. Therefore, we choose a random action. Random exploration is an important part of any deep RL algorithm. Therefore, in addition to this unlikely case, we add noise, $\mathcal{N}_k$, to the action, $u_k$, during the training. We reduce the amount of noise injected to the action as the algorithm converges.

---

**Algorithm 1** Locally Linear Q-Learning Training

---

1:  Initialize Q network (equation (15)) with random weights.
2:  Initialize target network, $Q'$, parameters: $\theta^{Q'} = \theta^Q$.
3:  Create the reply buffer $R = \emptyset$.
4:  **for** episode = 1:M **do**
5:      Initialize a random process $\mathcal{N}$ for action exploration.
6:      Receive the initial observation, $x_0$.
7:      **for** k = 1:T **do**
8:          **if** $||d_k|| \neq 0$ **then**
9:              Set $u_k = -(d_k^T d_k)^{-1}(d_k^T)h_k + \mathcal{N}_k$
10:         **else**
                Set $u_k = \mathcal{N}_k$
11:         Execute $u_k$ and observe $x_{k+1}$ and $r_k$.
12:         Store transition $(x_k, u_k, x_{k+1}, r_k)$ in $R$.
13:         **for** iteration = 1:$I_s$ **do**
14:             Randomly select a mini-batch of $N_s$ transition from $R$.
15:             Update $\theta^f$ and $\theta^g$ by minimizing the loss: $L_1 = \frac{1}{N_s}\sum_{i=1}^{N_s}||x_{i+1} - x_i - \Delta(f(x_i|\theta^f) + g(x_i|\theta^g)u_i)||$.
16:         **for** iteration = 1:$I_l$ **do**
17:             Randomly select a mini-batch of $N_l$ transition from $R$.
18:             Set $y_i = r_i + \gamma Q'(x_{i+1}|\theta^{Q'})$.
19:             Update $\theta^Q$ by minimizing the loss: $L_2 = \frac{1}{N_l}\sum_{i=1}^{N_l}||y_i - Q(x_i, u_i|\theta^Q)||$.
20:             Update the target network: $\theta^{Q'} = \tau\theta^Q + (1 - \tau)\theta^{Q'}$

---

In the application, the controller solves $u_k$ with additional constraints to achieve the desired short-term trajectories. We will discuss our short-term adjustment algorithms in the next section. To learn Q-function, in addition to the state estimation error, we minimize the long-term loss function, $L_2$, as it is presented in Algorithm 1. Note that having the short-term model, it is straightforward to add imagination rollout to our algorithm to increase sample efficiency. However, improving sample efficiency in RL is not the focus of this work.

## 4  CONTROL STRATEGY

By separating action design from prediction models, LLQL gives us the freedom to design different control strategies for achieving short-term and long-term goals. Moreover, the linear structure of short-term and long-term models simplifies the control design. Consider the case where LLQL has learned a perfect long-term model for an environment using Algorithm 1. In this case, the optimum

solution to achieve the long-term goal is given by equation (16). When we have one or more short-term goals as well, we can formulate the control design as an optimization problem to satisfy both short-term and long-term goals as much as possible.

In this paper, we consider two types of short-term goals: 1) desired trajectory, and 2) constraint. In the first scenario, the agent has a short-term desired trajectory. For example, a car may be required to travel with specific speed during certain periods. In the second scenario, the agent has some limitation for a specific period of time. For example, a car is required to keep its speed below certain thresholds at some periods during the trip. To address the first problem, we add an additional term to the cost function for the short-term goal and solve for the action. We deal with the second problem as a constraint optimization.

## 4.1 SHORT-TERM TRAJECTORY

Let $x_d$ represent our desired short-term trajectory. We develop a control strategy to track $x_d$ while pursuing the long-term goals.

Using system dynamic functions $f_k$ and $g_k$, we can define our control optimization problem as:

$$\min_{\text{find } u_k} (\gamma_1(h_k + d_k u_k)^2 + \gamma_2(x_{d(k+1)} - x_k - \Delta(f_k + g_k u_k))^2), \tag{17}$$

where $x_{d(k+1)}$ represents the desired trajectory at time $k+1$. $\gamma_1$ and $\gamma_2$ are positive coefficients and can be adjusted to give higher weights to the short-term or long-term goals. Note that in this work, we assume the short-term goals are temporary and when their time expires the system goes to the long-term optimum policy given by (16). For example, we may require a car to have a specific speed at some specific locations.

We can apply a similar pseudo-inverse matrix multiplication, and derive a solution with the least squares error for (17) as:

$$u_k^* = (\begin{bmatrix} \gamma_1 d_k \\ -\gamma_2 \Delta g_k \end{bmatrix}^T \begin{bmatrix} \gamma_1 d_k \\ -\gamma_2 \Delta g_k \end{bmatrix})^{-1} \begin{bmatrix} \gamma_1 d_k \\ -\gamma_2 \Delta g_k \end{bmatrix}^T \begin{bmatrix} -\gamma_1 h_k \\ \gamma_2(-x_{d(k+1)} + x_k + \Delta f_k) \end{bmatrix}. \tag{18}$$

## 4.2 SHORT-TERM CONSTRAINT

The LLQL algorithm provides a framework to design the actions considering different constraints. For safe operation, the agent may have to avoid specific states for a period of time (for example, high speed or locations close to an obstacle). For simplicity, we assume at each moment we only have maximum one constraint on one state variable, $x^i$. This is a reasonable assumption, because in physical systems the agent is close to one of the boundaries at any moment in time. When this is not the case, we can define new constraints as a combination of constraints. Consider $c_k^i$ as the constraint on the state variable, $x^i$, at time $k$. We can define the constraint optimization problem for LLQL as:

$$\min_{\text{find} u_k} \frac{1}{2}(h_k + d_k u_k)^2$$
$$\text{such that:} \tag{19}$$
$$x_{k+1}^i \le c_{k+1}^i.$$

$\frac{1}{2}$ is a coefficient added to simplify the mathematical operation. Using our estimation of the next step, $x_{k+1}^i = x_k^i + \Delta(f_k^i + g_k^i u_k)$, we can derive the optimum action which satisfies the constraint as:

$$u_k^* = -(d_k^T d_k)^{-1} d_k^T (h_k + \lambda^* \alpha_1), \tag{20}$$

where $\alpha_1 = \Delta g_k^i d_k^T (d_k d_k^T)^{-1}$, $\alpha_2 = \Delta g_k^i (d_k^T d_k)^{-1} d_k^T$, and $\lambda^* = \frac{x_k^i + \Delta f_k^i - c_{k+1}^i - \alpha_2 h_k}{\alpha_1 \alpha_2}$. The derivation details for short-term constraints are presented in Section C.

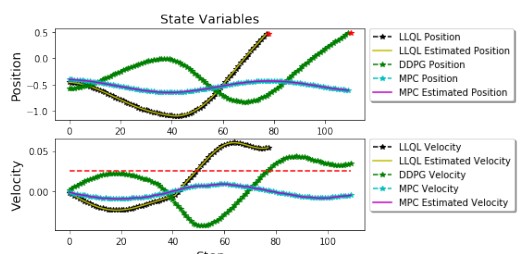

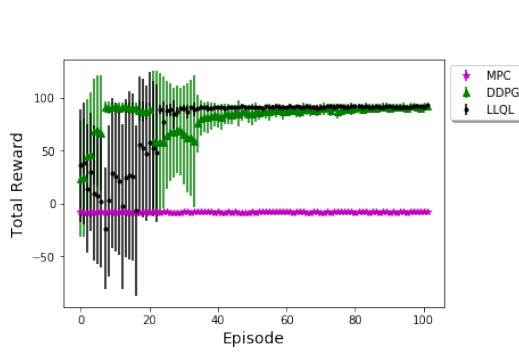

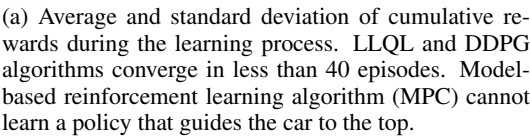

(b) State variables and estimated state variables. Using LLQL model, the car reached the top in 78 steps. Using DDPG model, the car reached the top in 110 steps. The car never reached the top when we used the MPC model. The mean absolute error for LLQL position estimation is 0.00078, the mean absolute error for LLQL velocity estimation is 0.000084, the mean absolute error for MPC position estimation is 0.00039 and the mean absolute error for MPC velocity estimation is 0.00014.

(a) Average and standard deviation of cumulative rewards during the learning process. LLQL and DDPG algorithms converge in less than 40 episodes. Model-based reinforcement learning algorithm (MPC) cannot learn a policy that guides the car to the top.

Figure 3: LLQL for MountainCarContinuous. The network's parameters are presented in Section D.

## 5 EXPERIMENTAL RESULTS

In this section, we demonstrate the performance of LLQL using Mountain Car with Continuous Action (MountainCarContinuous) from OpenAI Gym[1]. In Section B, we apply LLQL to control the crane system shown in Figure 1.

### 5.1 MOUNTAINCARCONTINUOUS

The MountainCarContinuous has two state variables: 1) car's position $-1.2 \leq x_k \leq 0.5$ and 2) car's velocity $-0.07 \leq v_k \leq 0.07$. $u_k$ is the continuous action at time $k$. A negative action pushes the car to the left and a positive action pushes the car to the right. The experience stops after $1,000$ steps or when the car reaches the goal on top of the mountain, $x_k = 0.5$, whichever occurs first. In the beginning of each episode, the car is randomly positioned at $-0.6 \leq x_0 \leq -0.4$. The reward for each episode is 100 for reaching the goal on top of the mountain minus the squared sum of actions from start to the goal. Figure 3a shows the cumulative rewards during the training for the LLQL, the DDPG, and a model-based reinforcement learning based on MPC presented by (Nagabandi et al., 2018). The details of the MPC based solution is presented in Section E. For each approach we performed the training 20 times, and selected the top 5 models with the maximum cumulative rewards to calculate mean and standard deviation of each episode in Figure 3a.

The MPC based solution uses the learned short-term predictive model (system dynamic model) to generate a sequence of actions that maximize the reward over a finite horizon. Figure 3b shows that the short-term predictive model estimates future states with high precision. However, optimizing for a finite horizon is a disadvantage for the model-based solution in achieving long-term goals. Increasing the horizon may improve the long-term performance, but it also increases the computational costs in the application phase. In our experiments, the car never reached the top of the mountain using the model-based method. Figure 3b presents the first 110 steps of a sample experiment. Unlike the MPC based solution, the LLQL and the DDPG algorithms converged in less than 40 episodes (see Figure 3a) and reach the top of the mountain in all experiments (see Figure 3b). Note that it is possible to improve the LLQL convergence time by applying imagination rollout. In fact, our short-term prediction model can be used to generate imaginary scenarios. However, sample efficiency is beyond the focus of this work. Figure 3 shows by applying the long-term predictive model, the LLQL algorithm outperforms the model-based reinforcement algorithm in achieving the long-term goals. In the next two subsections, we will show the LLQL algorithm can outperform the DDPG algorithm in achieving the short-term goals by using the short-term predictive model.

---

[1]http://gym.openai.com/envs/MountainCarContinuous-v0/

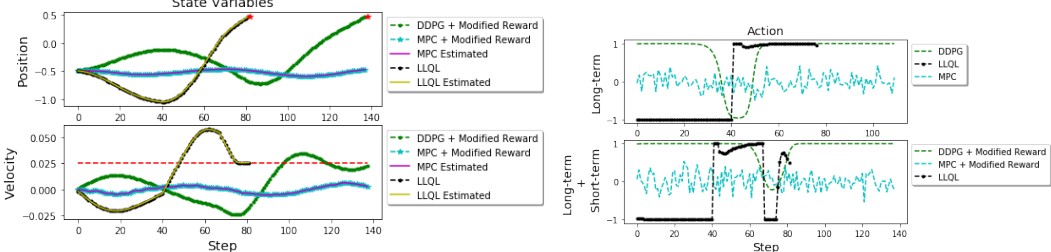

(a) With LLQL the car reached the top of the mountain in 82 steps with a short-term goal error of 0.00008. With DDPG + modified reward ($3^{rd}$ reward in Table 1) the car reached the top in 138 steps with short-term goal error 0.0023. The MPC + modified reward cannot guide the car to the top.

(b) Comparing the actions in normal case versus with the short-term trajectory. The hybrid strategy (see equation (21)) leads to quick adjustment in LLQL action and therefore, smaller short-term goal error compared to the DDPG + reward modification solution.

Figure 4: MountainCarContinuous with short-term and long-term goals. $\gamma_1 = 1$, $\gamma_2 = 2000$.

## 5.2 SHORT-TERM TRAJECTORY

Figure 3b shows that the policy presented in equation (16) can lead the car to the top of the mountain using the LLQL algorithm. We can see that the car's velocity is above 0.025 (the red line) when it reaches the top. Now consider the case where we want the car to reach the top of the mountain with our desired velocity, $v_d = 0.025$. Using equation (18), we can design a control strategy to reach this goal without requiring retraining the LLQL model. We apply the following hybrid control strategy to reach the top of the mountain with our desired speed.

$$u_k = \begin{cases} \text{use equation (16)} & \text{if } x_k < 0 \\ \text{use equation (18),} & \text{otherwise.} \end{cases} \tag{21}$$

Figure 4a shows that the car can reach the top of the mountain with our desired velocity. When we did not impose our desired speed to the system, the car reached the top of the mountain in 78 steps (see Figure 3b). Demanding a lower speed, slowed down the car and increased the number of steps to 82 (see Figure 4a). Figure 4b shows the actions with and without the short-term trajectory. We can see that the action temporarily becomes negative to reduce the velocity to the desired level and then goes back to positive to push the car to the top of the mountain.

To solve this problem in the traditional way, we had to modify the reward function to achieve both short-term and long-term goals. For comparison, we perform the following experience. We apply DDPG networks and MPC based networks with the modified reward functions shown in Table 1 to solve the MountainCarContinuous with the short-term and long-term goal problem. To achieve the short-term trajectory, we have considered different functions of the absolute error between the car's velocity and the desired velocity, $|v_d - v_k|$ in the final stages as an additional penalty. We have tried various architectures and hyper parameters to design the DDPG network. The final parameters are presented in Section D. We train a model for each modified reward function up to 300 episodes and save the model with maximum cumulative reward for our experiment. We run each model 10 times and measure the average number of steps it takes to reach the top of the mountain, and the average error between the car's final speed and the desired speed, $v_d = 0.025$. Table 1 shows that the MPC based solution cannot guide the car to the top. The DDPG with all the modified reward functions can achieve the long-term goal in all the experiences, reaching the top of the mountain in 10 out of 10 experiments. However, the DDPG based solutions do not perform very well with regard to the short-term goal. *On the other hand, LLQL does not require additional training or reward modification, achieves the long-term goal and has the least velocity error, 0.4%.*

## 5.3 SHORT-TERM CONSTRAINT

Now consider the case where it is unsafe to drive the car above a specific speed, for example, we plan to keep the speed under $v_k \leq 0.035$. We can use the following hybrid control strategy to achieve the

Table 1: Short-term trajectory performance

| RL Method | Modified Reward function | Average velocity error | Average number of steps | Long-term goal success rate |
|---|---|---|---|---|
| DDPG | $r_{new} = r_k - 5000 * \|v_k - v_d\|$ if done | 0.0232 | 109.1 | 10/10 |
| DDPG | $r_{new} = r_k - 100 * \|v_k - v_d\|$ if $x_k > 0.45$ | 0.0193 | 183.6 | 10/10 |
| DDPG | $r_{new} = r_k - 100 * \|v_k - v_d\|$ if $x_k > 0.45$ 
 $r_{new} = r_k - 5000 * \|v_k - v_d\|$ if done | 0.0088 | 173.8 | 10/10 |
| DDPG | $r_{new} = r_k - 25000 * (v_k - v_d)^2$ if done | 0.0193 | 103.3 | 10/10 |
| MPC | $r_{new} = r_k - 5000 * \|v_k - v_d\|$ if done | - | 1000 | 0/10 |
| MPC | $r_{new} = r_k - 100 * \|v_k - v_d\|$ if $x_k > 0.45$ | - | 1000 | 0/10 |
| MPC | $r_{new} = r_k - 100 * \|v_k - v_d\|$ if $x_k > 0.45$ 
 $r_{new} = r_k - 5000 * \|v_k - v_d\|$ if done | - | 1000 | 0/10 |
| MPC | $r_{new} = r_k - 25000 * (v_k - v_d)^2$ if done | - | 1000 | 0/10 |
| **LLQL** | - | **0.0001** | **89.5** | **10/10** |

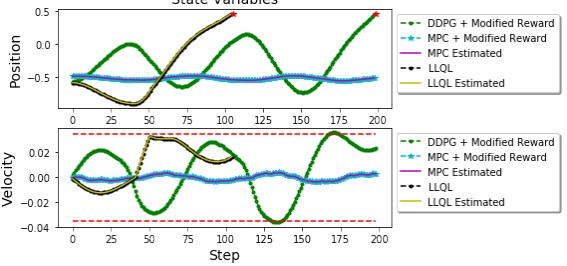 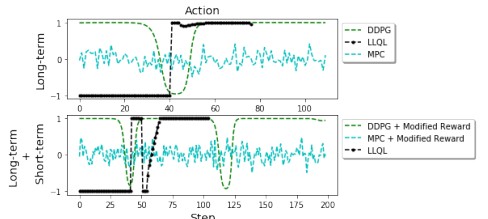

(a) With LLQL the car reached the top in 105 steps without violating the constraints. With DDPG + modified reward ($3^{rd}$ reward in Table 2) the car reached the top in 199 steps and violated the constraints 13 timesteps. The MPC + modified reward cannot guide the car to the top.

(b) Comparing the actions in the normal case versus with the short-term constraint. The hybrid strategy (see equation (22)) leads to sharp adjustment in LLQL action when the car gets close to the hazardous areas.

Figure 5: State variables for MountainCarContinuous with long-term goal and short-term constraint. The horizontal red lines, $|v_k| = 0.035$, represent the boundaries. The car reaches the goal in 97 steps.

long-term goal while keeping the speed safe:

$$u_k = \begin{cases} \text{use equation (16)} & \text{if } |v_k| \leq 0.033 \\ \text{use equation (20)}, & \text{otherwise.} \end{cases} \qquad (22)$$

We selected the boundary slightly less than the hazardous threshold (0.033 instead of 0.035) to be safe. Figure 5a shows that with the LLQL policy the car reaches its goal while staying outside of hazardous areas. The MPC based solution keeps the car outside of hazardous areas but cannot deliver the long-term goal (reaching the top of the mountain). The DDPG + modified reward reaches the top but fails to deliver the short-term goal (keeping the car out of hazardous areas). Like the previous section, we apply DDPG network + modified reward function and MPC + modified reward function to compare LLQL with the traditional model-free and model-based reward engineering approaches. We select the model with maximum cumulative rewards during 300 episodes of training. Table 2 shows that unlike LLQL, the modified rewards fail to keep the car below the allowed speed while reaching the top of the mountain. The model-based reinforcement learning algorithm baseline presented in Section E uses the same short-term network as the LLQL network. Table 1 and Table 2 show that even though the short-term part of our solution is useful in achieving short-term goals, it is not enough to solve the entire problem and achieve the long-term goal.

Table 2: Short-term constraint performance

| RL Method | Modified Reward function | Average number of steps out of boundary | Average number of steps | Long-term goal success rate |
|---|---|---|---|---|
| DDPG | $r_{new} = r_k - 10$ if $|v_k| > 0.033$ | 21.5 | 100 | 10/10 |
| DDPG | $r_{new} = r_k - 100(|v_k| - 0.033)$ if $|v_k| > 0.033$ | 25.2 | 106.8 | 10/10 |
| DDPG | $r_{new} = r_k - (100(|v_k| - 0.033))^2$ if $|v_k| > 0.033$ | 20.4 | 147.5 | 10/10 |
| DDPG | $r_{new} = -10$ if $|v_k| > 0.033$ | 25.1 | 104.1 | 10/10 |
| MPC | $r_{new} = r_k - 10$ if $|v_k| > 0.033$ | 0 | 1000 | 0/10 |
| MPC | $r_{new} = r_k - 100(|v_k| - 0.033)$ if $|v_k| > 0.033$ | 0 | 1000 | 0/10 |
| MPC | $r_{new} = r_k - (100(|v_k| - 0.033))^2$ if $|v_k| > 0.033$ | 0 | 1000 | 0/10 |
| MPC | $r_{new} = -10$ if $|v_k| > 0.033$ | 0 | 1000 | 0/10 |
| LLQL | - | **0** | **98.9** | **10/10** |

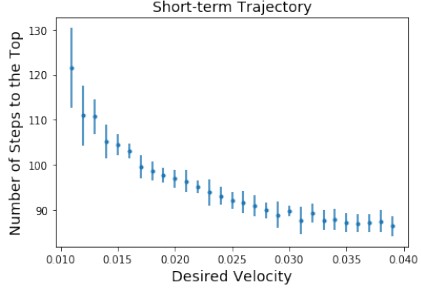

(a) Number of steps to the top vs desired final velocity. $\gamma_1 = 1$, $\gamma_2 = 2000$.

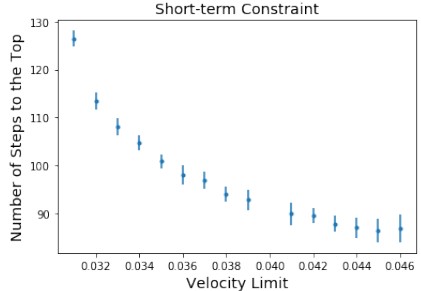

(b) Number of steps to the top vs velocity constraint.

Figure 6: Long-term performance vs short-term goals. We run the model with each short-term goal 10 times and present the average and standard deviation of the long-term goal.

## 5.4 EFFECT OF SHORT-TERM GOALS ON LONG-TERM PERFORMANCE

Using equations (18) or (20) for deriving a set of actions is equivalent to solving a sub-optimum solution for the long-term goal in order to satisfy the short-term desired trajectories or constraints. When the short-term goals are far from the global optimum solution, the long-term performance degrades. Figure 6a shows that lower desired velocities lead to longer traveling time for the Mountain-Car. Similarly, Figure 6b shows that further limiting the maximum velocity degrades the long-term performance.

## 6 CONCLUSIONS

In this work, we presented LLQL as a new model-based RL algorithm with the capability of achieving both short-term and long-term goals without requiring complex reward functions. By presenting the advantage function with a locally linear model and separating designing actions from the learning process, our method is capable of adopting control strategies to achieve different short-term goals without retraining the model. This can be very significant for industrial applications where the RL algorithms have not been used due to the necessity of different short-term adjustments.

The LLQL algorithm deviates from the optimal policy temporarily to address local short-term goals (trajectories or constraints). The agent would return to the optimum policy if the deviation is small enough that the agent is still in the environment explored during the training. In the future work, we will investigate conditions where short-term goals are feasible and develop a more analytical approach to set the meta parameters for the controller to guarantee short-term and long-term goals. Moreover, we will model uncertainties in short-term prediction model and apply robust control theory to design robust control solutions.

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

# A    RELATED WORK

Our work can be categorized as a new model-based RL approach. Model-based RL algorithms use the environment model which represents the state transition function to plan ahead and select actions that lead to higher rewards. Several model-based algorithms assume the environment model is known. Among them, *AlphaZero* (Silver et al., 2017) is one of the most famous. AlphaZero uses the game's rules (Chess, Shogi and Go) as the environment model to generate a series of self-play simulated games. During the simulations, the actions for both players are selected using a Monte-Carlo Tree Search (MCTS) algorithm. The MCTS performs as a planning algorithm by generating candidate actions which are superior to the current policy. The neural network parameters are updated at the end of each game to minimize the game prediction error (loss, draw or win) and maximize the similarity of policy vector to the planning algorithm. AlphaZero is limited to the discrete action space problems. The environment model is typically unknown in real-world applications. Therefore, many model-based RL algorithms learn the state transition model from the data. *NAF* (Gu et al., 2016) learns a linear model for state transition at each operating point and uses this model to generate additional samples through imagination rollout. *World Models* (Ha & Schmidhuber, 2018) uses a Variational Auto Encoder (VAE) to map a state variable, $x \in X$ to a lower dimensional variable $z$ in a latent space $Z$. It then uses a recurrent neural network (RNN) to learn the state transition model in the latent space. Finally, it applies a simple linear controller to $z$ and the hidden state in the RNN, $h$, to control the system.

Imagination-Augmented Agents (*I2As*) (Racanière et al., 2017) introduces two paths: 1) model-free path and 2) imagination path. The imagination path learns a transition model and uses this model to generate imagination rollouts. These rollouts are aggregated with the samples in the model-free path. To generate actions in the imagination path, I2As uses the model-free path policy. Therefore, the rollouts in the imagination path improve as the I2As policy improves. Using the imagination rollouts, I2As converge faster than a model-free network with the same number of parameters. Nagabandi et al. (2018) showed that a two-step control policy based on 1) learning the dynamic model and 2) applying MPC to the learned model is significantly more sample efficient than model-free RL. However, this approach cannot achieve high rewards. To achieve higher rewards and preserve sample efficiency, they proposed a hybrid model-based and model-free (*MBMF*) algorithm which runs the model-based approach to achieve the initial result in a sample efficient way, it then trains a model-free policy to mimic the learned model-based controller, and uses the resulting imitation policy as the initialization for the final model-free RL algorithm.

Feinberg et al. (2018) proposed Model-based Value Expansion (*MVE*) algorithm, which limits the uncertainty in the model by only allowing imagination up to a fixed number of steps, H. MVE uses the learned system dynamic model to generate simulation data up to H steps into the future, and applies these sample points to estimate Q-function. Instead of saving simulated samples in an imagination buffer, MVE retrains the dynamic model and generates a new imagination rollout at each step. Buckman et al. (2018) expanded MVE algorithm by proposing Stochastic Ensemble Value Expansion (*STEVE*), to generate a solution more robust to model uncertainty. Dalal et al. (2018) proposed safe exploration by modeling constraints using a linear model and applied Lagrangian optimization to modify the action in order to guarantee safety. In this work, we also used Lagrangian optimization for short-term constraints. However, our approach is different in two ways: 1) our method does not modify the RL action to achieve the goals. Instead, it derives an action by considering both long-term goals and short-term constraints. This is possible because our algorithm uses a locally linear model to represent the advantage function. 2) Unlike safe exploration, the focus of this paper is in handling new constraints in the application phase without retraining the model.

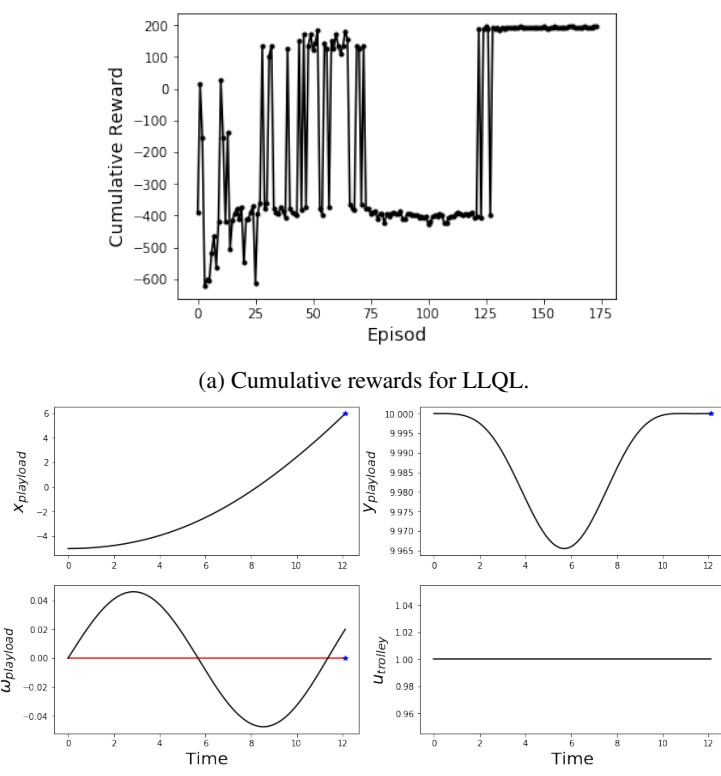

(a) Cumulative rewards for LLQL.

(b) State variables for the crane system. The crane reaches the goal in 12s.

Figure 7: LLQL for the crane system.

## B    CRANE CONTROL SYSTEM

Gantry cranes are widely used in industrial production lines and construction projects for transferring heavy and hazardous materials. The objective of the crane system is to convey the payload from the start position to the destination position as soon as possible while keeping the payload sway at the destination minimum. Higher traveling speed improves the efficiency and reduces costs. However, excessive movements at the destination wastes time and energy and can lead to accidents. To move the payload as fast as possible and stop the sway at the destination, skillful operators are required. Labor shortage in industries, and risk of human error, have motivated us to develop an automated solution for crane control. The crane dynamic system is highly nonlinear. Traditional nonlinear control techniques such as sliding control (Bartolini et al., 2002) and adaptive control (Boustany & d'Andrea Novel, 1992) have been applied to these systems. These methods require detailed mathematical model of the system and its environment, which can be complicated and expensive to derive. When a simulator is available for a crane system, RL algorithms can provide a compelling alternative to traditional control methodologies. This is the case in many industries, where for intellectual property concerns the companies are willing to provide simulators to the costumers but refuse to reveal mathematical models of their products.

Our crane simulator provides us six state variables: 1) trolley location, $x_{trolley}$ 2) trolley velocity, $v_{trolley}$, 3) payload angle, $\phi_{payload}$, 4) payload angular velocity, $\omega_{payload}$, 5) payload horizontal location, $x_{payload}$, and 6) payload vertical location, $y_{payload}$. The only action is the force applied to the trolley, $u_{trolley}$. The overall goal is to reach the final destination $x_{pd}$ and $y_{pd}$ in the shortest time possible. We choose the following reward function to learn a policy to do so.

$$r_k = \begin{cases} 500 & \text{if } |x_{payload}(k) - x_{pd}| < \epsilon \ \& \ |y_{payload}(k) - y_{pd}| < \epsilon \\ -1 & \text{otherwise,} \end{cases} \qquad (23)$$

where $\epsilon$ is a small constant. Figure 7b shows that our learned policy pushes the trolley with maximum force, $u_{trolley}$ is eqaual 1 for the entire episode, till the payload reaches the goal $x_{pd} = 6, y_{pd} = 10$.

In additional to the long-term goal, our short-term goal is to minimize the object's sway when it reaches to the final destination. Instead of designing complicated reward functions to achieve minimum travel time and minimum sway, we consider $\omega_{payload} = 0$ at the final destination as a short-term desired trajectory. We consider the following hybrid strategy to reach the final destination with close to zero sway.

$$u_{trolley} = \begin{cases} \text{use equation (16)} & \text{if } x_{payload} < 5.5 \\ \text{use equation (18)}, & \text{otherwise} \end{cases} \tag{24}$$

Figure 8 shows that our strategy can reach the destination with close to zero swing.

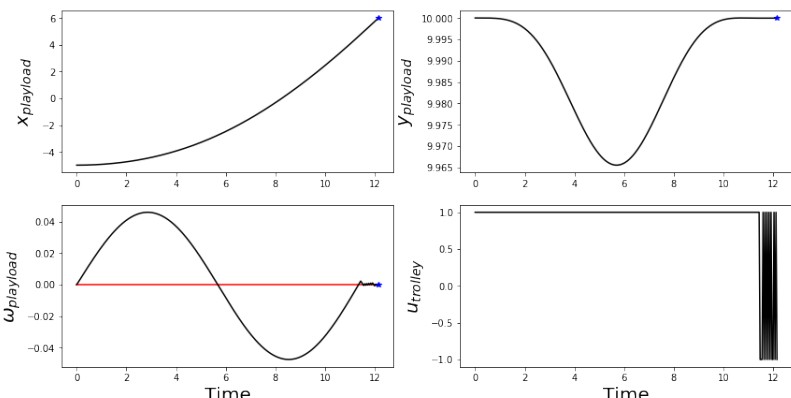

Figure 8: State variables for the crane system with required short-term trajectory, $\omega_{payload} = 0$. The final error is 0.0001. The crane reaches the goal in 12.5s. $\gamma_1 = 1, \gamma_2 = 1000$.

## C   DERIVATION DETAILS FOR SHORT-TERM CONSTRAINTS

Consider equation (19). Substituting our estimation of the next step from equation (14) in (19), we have

$$
\begin{aligned}
&\min_{\text{find } u_k} \frac{1}{2}(h_k + d_k u_k)^2 \\
&\text{such that:} \\
&x_k^i + \Delta(f_k^i + g_k^i u_k) \le c_{k+1}^i
\end{aligned}
\tag{25}
$$

Using Lagrangian method at each time step $k$, we have

$$
\begin{aligned}
L(u_k, \lambda) = &\frac{1}{2}(h_k + d_k u_k)^2 + \\
&\lambda(x_k^i + \Delta(f_k^i + g_k^i u_k) - c_{k+1}^i)
\end{aligned}
\tag{26}
$$

Taking the gradient of $L$ with respect to $u_k$, we can write the Karush-Kuhn-Tucker (KKT) (Kuhn & Tucker, 2014) conditions for optimal solution of equation (25), $\{u_k^*, \lambda^*\}$ as:

$$
\begin{aligned}
&(h_k + d_k u_k^*)d_k + \lambda^* \Delta g_k^i = 0 \\
&\lambda^*(x_k^i + \Delta(f_k^i + g_k^i u_k^*) - c_{k+1}) = 0
\end{aligned}
\tag{27}
$$

With this assumption, we can show

$$
u_k^* = -(d_k^T d_k)^{-1} d_k^T (h_k + \lambda^* \Delta g_k^i d_k^T (d_k d_k^T)^{-1})
\tag{28}
$$

Note that when there is no constraint: $\lambda^* = 0$, we have $u_k^* = -(d_k^T d_k)^{-1} d_k^T h_k$. This is exactly the input we computed in equation (16). When $\lambda^* \ne 0$, we have

$$
x_k^i + \Delta(f_k^i + g_k^i u_k^*) - c_{k+1} = 0
\tag{29}
$$

We define $\alpha_1 = \Delta g_k^i d_k^T (d_k d_k^T)^{-1}$, and $\alpha_2 = \Delta g_k^i (d_k^T d_k)^{-1} d_k^T$. $\alpha_1$ and $\alpha_2$ are scalar. Substituting $u_k^*$ from equation (28) in equation (29) we have

$$
x_k^i + \Delta f_k^i - \alpha_2(h_k + \alpha_1 \lambda^*) - c_{k+1} = 0
\tag{30}
$$

Therefore,

$$
\begin{aligned}
\lambda^* &= \frac{x_k^i + \Delta f_k^i - c_{k+1} - \alpha_2 h_k}{\alpha_1 \alpha_2} \\
u_k^* &= -(d_k^T d_k)^{-1} d_k^T (h_k + \lambda^* \alpha_1).
\end{aligned}
\tag{31}
$$

## D    NETWORKS PARAMETERS

We used the following network structures and parameters in the experimental studies.

*LLQL for MountainCarContinuous:* $h$, $d$, $V$, $f$ and $g$ networks each has two hidden layers with 200 neurons in each layer. All the activation functions are Rectified Linear Units (ReLUs). Each episode is maximum 1000 steps. The number of iterations for short-term and long-term prediction model: $I_s = I_l = 5$. The learning rate for the long-term prediction model is $0.001$. The batch size for this model is 10. The discount rate $\gamma = 0.999$. The target model update rate, $\tau = 0.001$. The learning rate for the short-term prediction model is $0.001$ for the first 20000 steps and then reduces to $0.0001$. $\Delta = 0.001$. The batch size for this model is 100.

*LLQL for the crane system:* $h$, $d$, $V$, $f$ and $g$ networks each has two hidden layers with 200 neurons in each layer. All the activation functions are ReLUs. Each episode includes maximum 1000 actions. The number of iterations for short-term and long-term prediction model: $I_s = I_l = 5$. The learning rate for the long-term prediction model is $0.001$. The batch size for this model is 10. The discount rate $\gamma = 0.999$. The target model update rate, $\tau = 0.001$. The learning rate for the short-term prediction model is $0.01$ for the first 20000 steps and then reduces to $0.001$. $\Delta = 0.001$. The batch size for this model is 200.

*DDPG networks with modified reward functions:* The Q-network, and the deterministic policy network each has two hidden layers with 200 neurons. All the activation functions are ReLUs. The learning rate for the Q-network is $0.00001$, and the learning rate for the deterministic policy network is $0.000001$. The discount rate $\gamma = 0.99$. The batch size is 8. The target model update rate for both networks is $0.1$.

In all the networks we shift and scale the state variables to zero mean and unit standard deviation for a better learning. For exploration, we add an additive normal noise to the action:

$$u(k) = (1 - \alpha_N)u(k)^* + \alpha_N \mathcal{N}, \tag{32}$$

where $u(k)^*$ represents the optimum action generated by LLQL or DDPG, and $\mathcal{N} = U(-1, 1)$ is a continuous uniform random variable. We consider $\alpha_N = 0.05$ in the beginning, and $\alpha_N = .99 \times \alpha_N$ after each episode with positive cumulative rewards.

# E    MODEL-BASED REINFORCEMENT LEARNING

Nagabandi et al. (2018) proposed a model-based RL based on MPC. Their approach uses the system dynamic model + reward function to generate a sequence of actions that maximize the cumulative reward. To represent the effect of short-term and long-term predictive models in our algorithm, we present a modified version of (Nagabandi et al., 2018) algorithm using our short-term predictive model, $f$ and $g$ in equation (14). The model based RL uses the learned dynamic model, $f$ and $g$ networks, and generates a sequence of actions $U_k^H = (u_k, ..., u_{k+H-1})$ to maximize reward over a finite horizon, $H$.

$$U_k^H = \arg\max \sum_{i=k}^{k+H-1} r(\hat{x}_i, u_i),$$

(33)

$$\text{such that: } \hat{x}_k = x_k \text{ and } \hat{x}_{i+1} = \hat{x}_i + \Delta(f(\hat{x}_i|\theta^f) + g(\hat{x}_i|\theta^g)u_i)$$

Solving equation (33) for the exact solution is computationally expensive. Therefore, Nagabandi et al. (2018) applied a simple random sampling method and select the candidate action sequence with the highest expected cumulative reward. In this work, we apply the same method to solve equation (33). We set $H = 25$ and select 10000 samples at each step. Following the MPC closed-loop control framework, the algorithm only executes the first action in the sequence and calculate a new sequence in the next step. Algorithm 2 presents the model-based RL.

---

**Algorithm 2** Model-based Reinforcement Learning (Nagabandi et al., 2018)

---

1: Initialize $f$ ang $g$ networks (equation (14)) with random weights.
2: Create dataset of random trajectories $R_{rand}$.
3: Create the reply buffer $R_{rl} = \emptyset$.
4: **for** episode = 1:M **do**
5:     Randomly select a mini-batch of $N_s$ transition from $R_{rand}$ and $R_{rl}$.
6:     Update $\theta^f$ and $\theta^g$ by minimizing the loss: $L_1 = \frac{1}{N_s} \sum_{i=1}^{N_s} ||x_{i+1} - x_i - \Delta(f(x_i|\theta^f) + g(x_i|\theta^g)u_i)||$.
7:     **for** k = 1:T **do**
8:         Solve equation (33) for the optimum sequence $U_k^H$ at state $x_k$.
9:         Executes the first action in the sequence, $u_k$.
10:        Store transition $(x_k, u_k, x_{k+1}, r_k)$ in $R_{rl}$.

---

