# OpenReview forum: "Long-term planning, short-term adjustments"
_ICLR.cc/2020/Conference — Reject_

### Official Review · AnonReviewer2 · 2019-10-22
**Official Blind Review #2**

**Rating:** 3

**Review:**

The paper proposes a method for combining continuous Q learning and linear MPC. They propose learning a continuous control policy for long term behavior trained with standard model free reinforcement learning and a short term linear control model implemented as a linear dynamical system. They propose learning them separately each with its own loss function. At test time a combined controller can be used to infer actions by solving small optimization problem. The linearity makes inference very efficient and even when additional constraints are added only at test time. The model should therefore me more flexible and allows for adapting 0 shot when the constraints are known. The paper is well written and mostly clear.

The equations seem correct but I am not an expert in continuous control. The examples considered seem a bit too simple to be insightful.

* the algorithm seems to be be setup to with the dynamics model being learnt on the transitions coming from an epsilon greedy with respect to the optimal policy. Does this mean that generalization may be affected if the model strays off the optimal trajectories ?
* There would be some benefit in trying the model in some other domain maybe reacher with some constraints ?



**Experience Assessment:**

I have read many papers in this area.

**Review Assessment: Checking Correctness Of Derivations And Theory:**

I assessed the sensibility of the derivations and theory.

**Review Assessment: Checking Correctness Of Experiments:**

I assessed the sensibility of the experiments.

**Review Assessment: Thoroughness In Paper Reading:**

I made a quick assessment of this paper.

---

> ### Author Response · Authors · 2019-11-15
> **We appreciate the reviewer’s comments. We revised the paper based the comments as follows.**
>
> 1- This is a very good point. The LLQL algorithm deviates from the optimal policy temporarily to address local short-term goals (trajectories or constraints). The agent would return to the optimum policy if the deviation is small enough that the agent is still in the environment explored during the training.  We have clarified this point in the conclusion as:
> “The LLQL algorithm deviates from the optimal policy temporarily to address local short-term goals (trajectories or constraints). The agent would return to the optimum policy if the deviation is small enough that the agent is still in the environment explored during the training. In the future work, we will investigate conditions where short-term goals are feasible and develop a more analytical approach to set the meta parameters for the controller to guarantee short-term and long-term goals. Moreover, we will model uncertainties in short-term prediction model and apply robust control theory to design robust control solutions.”
>
> 2- We have extended the experiment to compare our algorithm with a model-based reinforcement learning in addition to the model-free bassline, DDPG. We also updated Figures 3-5 to further present the differences between the LLQL and the two baselines. Moreover, we have presented a second case study based on a simulator developed for real world applications in our company, and based on the LLQL promising results, we plan to apply it in the operation. We had to put the second case study (the crane system) in the appendix as we were out of space and we did not want to cut more important theoretical parts of the paper.

---

### Official Review · AnonReviewer3 · 2019-10-26
**Official Blind Review #3**

**Rating:** 6

**Review:**

Thank the authors for the response. I will keep my score.
----------------------------------------
Summary
In this paper, the author proposes LLQL (Locally Linear Q-Learning), which separates the action design from prediction models, such that the model can have short-term and long-term goals. The short-term network takes the current state and the action, and predicts the next state (to be specific, the status change) under the assumption that the output is linear to the action. The long-term network takes the current state as input, controls the controller to take action, and also optimizes a Q-function. I lean to vote for accepting this paper, though the experiment part can be further improved.
Strengths
- The proposed model is novel, which can take a trajectory or a constraint as a short-term goal. As a result, we can control the behavior of the model easily. Otherwise, we have to design a new reward function to guide our model indirectly. As the authors showed in Tables 1 & 2, LLQL outperforms this approach.
- The paper is clearly written. The author provides clean formulas throughout the paper, which makes the paper easy to understand.
Weaknesses
More experiments can be conducted to demonstrate the effectiveness of LLQL.
- Only tested in one scenario. In the main text, though it is already very long, there are still only experiments in one scenario (i.e., Mountain Car).
- Only compared to one baseline. In order to demonstrate the effectiveness of LLQL taking a short-term trajectory or constraint, the authors compared it to only one method (i.e., DDPG) with a few manually designed reward function.
Minor
Equation 6 looks incorrect to me.
Possible Improvements
As mentioned before, it would be great to include more experiments in the main text, comparing LLQL to more baselines, testing LLQL in more scenarios, or doing some ablation studies (removing one part of LLQL to demonstrates the effectiveness of that part).

**Experience Assessment:**

I have read many papers in this area.

**Review Assessment: Checking Correctness Of Derivations And Theory:**

I assessed the sensibility of the derivations and theory.

**Review Assessment: Checking Correctness Of Experiments:**

I assessed the sensibility of the experiments.

**Review Assessment: Thoroughness In Paper Reading:**

I read the paper at least twice and used my best judgement in assessing the paper.

---

> ### Author Response · Authors · 2019-11-15
> **We would like to thank you the reviewer for very positive and encouraging comments. We modified the paper based the comments as follows.**
>
> 1- We have added an experiment to compare our algorithm with a model-based reinforcement learning in addition to the model-free baseline, DDPG. We also updated Figures 3-5 to further present the differences between the LLQL and the two baselines. These figures plus Table 1 and Table 2 show that the model-based reinforcement learning baseline struggles to achieve the long-term goal as for the mountain car immediate gains does not guarantee long-term success, and the model-free reinforcement learning baseline cannot deliver short-term goal as they are hard to be captured in a solution highly rewarded by the long-term goal.  The LLQL uses both the system dynamic model (short-term predictive model) and the Q-learning (long-term predictive model), to achieve both short-term and long-term goal.
>
> The second case study in the paper (the crane system) is based on a simulator developed for real world applications in our company, and based the LLQL promising results, we plan to apply it in the operation. We had to put the crane case study in the appendix as we were out of space and we did not want to cut more important theoretical parts of the paper.
>
> 2- We have fixed equation (6).
>
>
> 3-We appreciate the reviewer suggestion for removing one part of the LLQL to demonstrates the effectiveness of that part. Inspired by this comment, we designed the model-based reinforcement learning algorithm baseline to use the same short-term network as the LLQL. This additional experiment showed that even though the short-term part of our solution is useful in predicting short-term future and achieving short-term goals, it is not enough to solve the entire problem and achieve the long-term goal as well. We have expressed this in the paper as:
> “The model-based reinforcement learning algorithm baseline presented in Section E uses the same short-term network as the LLQL network. Table 1 and Table 2 show that even though the short-term part of our solution is useful in achieving short-term goals, it is not enough to solve the entire problem and achieve the long-term goal.”

---

### Official Review · AnonReviewer1 · 2019-10-28
**Official Blind Review #1**

**Rating:** 3

**Review:**

This paper proposes a Locally Linear Q-Learning (LLQL) method for continuous action control. It uses a short-term prediction model and a long-term prediction model to generate actions that achieve short-term and long-term goals simultaneously. The problem the paper seeks to solve is important. However, this paper has several issues.

First, there seems to be an over-claim of the contribution. The proposed method is more like hybrid of model-based and model-free RL method. Specifically, the short-term prediction model is in fact the linearized dynamic system with system parameters modeled by deep neural networks, while the long-term prediction model is in fact different (state- and action-) value functions. For this reason, it is probably unnecessary to name them as “short-term network” or “long-term network”, since they are simply the system model (or the model-based part) and the value functions (the model-free part).

Second, the proposed method is not sufficiently evaluated. It is only evaluated on the toy Mountain Car task (and the Crane system in supplementary material). In order to justify the performance of an RL algorithm for continuous action space, it should be at least evaluated on the set of MuJoCo tasks.

Third, the proposed method is not sufficiently compared with different baselines. In Figures 3-5, the proposed LLQL algorithm is never compared to any baseline method, leaving it open whether it is actually better than earlier methods like DDPG. In Table 1, LLQL is compared to DDPG (a model-free method), and is shown to achieve better performance. However, this seems to be unfair because the proposed method is in fact a model-based RL algorithm. Therefore, it should at least compare to other model-based algorithms (and also other riche set of safe-exploration RL methods).

Other comments:
•	In eqn. (6), \gamma should be \gamma^{i-k}?
•	In the paragraph after (8), “Q-learning algorithms (16)…” is referring to a wrong equation (16) for Q-learning. Or probably the authors are not using the correct format to cite the reference. (This seems to happen repeatedly in later part of the paper such as as in the paragraph between (14) and (15).) It confuses the equation number and the reference number.
•	More explanation should be given about d(x_k|\theta^d) and h(x_k|\theta^h) after (15). The meaning of them has never to defined before.


**Experience Assessment:**

I have published in this field for several years.

**Review Assessment: Checking Correctness Of Derivations And Theory:**

N/A

**Review Assessment: Checking Correctness Of Experiments:**

I assessed the sensibility of the experiments.

**Review Assessment: Thoroughness In Paper Reading:**

I read the paper thoroughly.

---

> ### Author Response · Authors · 2019-11-15
> **We appreciate the reviewer’s comments. We modified the paper based the comments as follows.**
>
> 1- We agree with the reviewer that the short-term model is our estimation of the system dynamic, and the long-term model is our estimation of value function + advantage function. We name these models in our approach “short-term prediction model” and “long-term prediction model” to emphasize on their role in our hybrid solution, which are realized as deep neural networks. With the purpose of naming each model with the functionalities, we do not aim to claim they are novel networks. The novelty and contribution of our work is to propose a new framework which combines the system dynamic model (short-term predictive model) and the Q-learning (long-term predictive model) to achieve short-term and long-term goals simultaneously. We clarified this point in the paper as
>  “In this paper, we present a Locally Linear Q-Learning (LLQL) algorithm for continuous action space. The LLQL includes a short-term prediction model, a long-term prediction model, and a controller. The short-term prediction model represents a locally linear model of the dynamic system, while the long-term prediction model represents the value function + a locally linear advantage function. The controller uses the short-term prediction model and the long-term prediction model to generate actions that achieve short-term and long-term goals simultaneously. It adopts a policy that maximizes Q-value while achieving short-term goals. The LLQL algorithm has the following advantages:
> • It does not require designing sensitive reward functions for achieving short-term and long-term goals concurrently.
> • It shows better performance in achieving short-term and long-term goals compared to the traditional reward modification methods.
> • It is possible to modify the short-term goals without time-consuming retraining.”
>
> 2-We chose the Mountain Car to evaluate our approach because it clearly shows the necessity of considering long-term goals as the car must go in opposite direction in the beginning to gain enough momentum to reach the top of the mountain. In many other examples, a simple MPC method which only considers short-term gains can also deliver long-term goals.  We added an experiment based on model-based reinforcement learning to show this in the revised version of the paper.
> The second case study in the paper (the crane system) is based on a simulator developed for real world applications in our company, and based the LLQL promising results, we plan to apply it in the operation. We had to put the crane case study in the appendix as we were out of space and we did not want to cut more important theoretical parts of the paper.
> The motivation of our work comes from industrial applications such as crane control, where the emphasis is not exactly the same as cases such as MuJoCo where the most interesting cases are often in high DoFs and complicated system dynamics, but relatively simpler goal. In our cases, we aim to have complicated goals while the system dynamics are relatively straightforward. We agree that it will be interesting to apply our proposed method to MuJoCo environment where both the system and the goal are complicated. Unfortunately, when we were writing the paper our company did not have purchased the commercial licenses for MuJoCo, and therefore, we couldn’t test our algorithm on MuJoCo tasks. We have started the process to gain the license to further test and validate our solution.
>
> 3- We have updated Figure 3-5 to include DDPG as a model-free reinforcement learning baseline, and a model-based reinforcement learning baseline presented in (Nagabandi et al 2018). We also updated Table 1 and Table 2 to include the model-based RL baseline.
> The model-based reinforcement learning baseline struggles to achieve the long-term goal as for the mountain car immediate gains does not guarantee long-term success, and the model-free reinforcement learning baseline cannot deliver short-term goal as they are hard to be captured in a solution highly rewarded by the long-term goal.  The advantage of the LLQL is that by using both the system dynamic model (short-term predictive model) and the Q-learning (long-term predictive model), it can bring us the best of both worlds.
> Our solution algorithm is fundamentally different with safe-exploration problems, as they are designed for the training phase to guarantee safe exploration, but our proposed solution is applied during the application without retraining the model.
>
> 4- We have fixed equation (6).
>
> 5-We have fixed the references based on the conference template.

---

> > ### Author Response · Authors · 2019-11-15
> > **Remainder of our response**
> >
> > 6- h(xk|\theta^h) and d(xk|\theta^d) represents a linearized model for the advantage function. We had to model the advantage function, and the dynamic system using locally linear models, so we can solve a simple optimization problem for an action that can satisfy both short-term and long-term goals.
> > We clarified this point in the paper as:
> > “However, we present the advantage function, A(x, u|theta^A)  using a locally linear function of xk and uk as:
> > A(x, u| theta^A) = ||(h(xk| theta^h) + d(xk| theta^d)uk)||,
> >  (15)
> > where h(xk|\theta^h) and d(xk|\theta^d) networks model the locally linear advantage function.”

---

### Decision · Program_Chairs · 2019-12-19

**Decision:**

Reject

**Comment:**

This paper proposes a reinforcement learning algorithm for continuous action domains that combines a short-horizon model-based objective and a long-horizon value estimate.  The stated benefits to this approach are the ability to modify the model-based objective without extensive retraining.  This model-based objective can also capture custom constraints and implements a linear dynamics model that is used in conventional control theory.  The proposed method was evaluated on two domains (the mountain car domain and a custom crane domain) and compared to a continuous action space method (DDPG).

This discussion of this paper highlighted both strengths and weaknesses.  The reviewers said the presentation was clear.  The reviewers also appreciated the relevance of the problem.  The primary weakness was the evaluation of the method. One repeated concern from the reviewers was having only one standard domain for evaluation (mountain car).  Another concern was the absence of other model-based algorithms, which was addressed by the author response.

This paper is not yet ready to be published, despite its possible benefits,  due to the lack of evidence for this method on more continuous action problems.